# EGFR-Targeted Perfluorohexane Nanodroplets for Molecular Ultrasound Imaging

**DOI:** 10.3390/nano12132251

**Published:** 2022-06-30

**Authors:** Sidhartha Jandhyala, Austin Van Namen, Catalina-Paula Spatarelu, Geoffrey P. Luke

**Affiliations:** 1Thayer School of Engineering, Dartmouth College, Hanover, NH 03755, USA; sjandhy.th@dartmouth.edu (S.J.); austin.c.van.namen.th@dartmouth.edu (A.V.N.); catalina-paula.spatarelu.th@dartmouth.edu (C.-P.S.); 2Translational Engineering in Cancer Program, Dartmouth Cancer Center, Lebanon, NH 03756, USA

**Keywords:** perfluorocarbon nanodroplet, ultrasound imaging, molecular targeting, acoustic droplet vaporization

## Abstract

Perfluorocarbon nanodroplets offer an alternative to gaseous microbubbles as contrast agents for ultrasound imaging. They can be acoustically activated to induce a liquid-to-gas phase transition and provide contrast in ultrasound images. In this study, we demonstrate a new strategy to synthesize antibody-conjugated perfluorohexane nanodroplet (PFHnD-Ab) ultrasound contrast agents that target cells overexpressing the epidermal growth factor receptor (EGFR). The perfluorohexane nanodroplets (PFHnD) containing a lipophilic DiD fluorescent dye were synthesized using a phospholipid shell. Antibodies were conjugated to the surface through a hydrazide-aldehyde reaction. Cellular binding was confirmed using fluorescence microscopy; the DiD fluorescence signal of the PFHnD-Ab was 5.63× and 6× greater than the fluorescence signal in the case of non-targeted PFHnDs and the EGFR blocking control, respectively. Cells were imaged in tissue-mimicking phantoms using a custom ultrasound imaging setup consisting of a high-intensity focused ultrasound transducer and linear array imaging transducer. Cells with conjugated PFHnD-Abs exhibited a significantly higher (*p* < 0.001) increase in ultrasound amplitude compared to cells with non-targeted PFHnDs and cells exposed to free antibody before the addition of PFHnD-Abs. The developed nanodroplets show potential to augment the use of ultrasound in molecular imaging cancer diagnostics.

## 1. Introduction

Cancer diagnostics using molecular imaging have been explored as a less invasive method to determine disease state and progression. Molecular imaging involves an imaging modality, a contrast agent, and a targeting moiety. Medical imaging modalities such as positive emission tomography (PET) [1], computed tomography (CT) [2], magnetic resonance imaging (MRI) [3] and ultrasound imaging [4] have been used in molecular imaging applications, each with their own advantages and disadvantages. Molecular imaging using PET is highly sensitive and has great penetration depth, but requires radionuclide contrast agents, and has low spatial resolution [5]. CT allows for deep tissue penetration and high spatial resolution, but exposes patients to ionizing radiation [5]. MRI provides detailed images without the use of ionizing radiation, but is time and cost intensive [5]. Ultrasound has good temporal resolution, is inexpensive and does not require radiation or nuclides; however, commercially available contrast agents are large and do not diffuse into tissue easily [5]. Developing smaller ultrasound molecular imaging agents could augment the modality’s potential for cancer diagnostics.

Ultrasound microbubble contrast agents have been used in various diagnostic and therapeutic applications, such as vascular mapping, tissue ablation, clot disruption, targeted drug delivery, and lithotripsy [6,7,8,9]. Typically, microbubbles consisting of air or gaseous perfluorocarbon cores are used to provide imaging contrast due to the large acoustic impedance mismatch between their gaseous core and the surrounding tissue. While microbubbles have proven to be effective contrast agents, their relatively large size (mean diameter > 1 µm) restricts them to the vasculature and results in rapid clearance from the bloodstream [10].

Nano-sized contrast agents overcome the size restrictions of microbubbles by allowing for permeation into leaky vasculature of tumors, leading to potential ultrasound-guided tumor-specific molecular imaging targets [11]. One particular class of ultrasound contrast agents, perfluorocarbon nanodroplets (PFCnDs), have been studied due to their phase change properties [12]. The PFCnDs contain a liquid perfluorocarbon core surrounded by a lipid, protein, or polymer shell. The ultrasound contrast is produced when the PFCnDs are activated using an optical or acoustic stimulus to induce the liquid perfluorocarbon core to vaporize into a gaseous microbubble [13,14]. The boiling point of the perfluorocarbon core can be either below or above the temperature of the surrounding tissue. In the former case, the nanodroplets can be activated with a relatively low level of energy to undergo a single liquid-to-gas transition. In the latter case, a higher energy level is required to initiate vaporization, and the resulting transient microbubble recondenses back to the original nanodroplet form after several milliseconds [15,16]. The vaporization/recondensation process can be repeated multiple times. Thus, the higher-boiling-point core yields the benefit of improved stability and controllable contrast. The phase change dynamics of the PFCnDs have led to their use in a variety of imaging and therapeutic applications [15,17,18,19,20].

Perfluorocarbon nanodroplets have the potential to enhance ultrasound imaging by providing molecular information through various targeting strategies. Multiple publications have reported the development of perfluorocarbon nanodroplets to target extracellular markers such as folate receptors [21,22], and human epidermal growth factor receptor-2 (HER2) [23,24]. While these examples show targeting to specific molecules, the perfluorocarbon core used in the droplets have boiling points lower than the surrounding tissue, leading to one-time activation and instability.

In this paper, we describe epidermal growth factor receptor (EGFR) antibody-functionalized phase-change perfluorohexane core nanodroplets (PFHnD-Ab) for molecular ultrasound imaging. The EGFR was chosen as the target due to its over-expression in over 90% of head and neck squamous cell carcinomas (HNSCC) [25,26]. The perfluorohexane core has a boiling point of 56 °C, allowing for repeated vaporization and recondensation of the PFHnD-Ab, enabling more robust imaging. The molecular specificity is conferred through directional attachment of EGFR antibodies to the surface of the nanodroplets through a hydrazide-aldehyde reaction. The targeting specificity of the PFHnD-Abs was demonstrated through fluorescence microscopy and ultrasound imaging [16]. The impact of repeated vaporization cycles on molecular binding was investigated and the cellular imaging sensitivity was determined in-vitro. Overall, the PFHnD-Ab have the potential to expand ultrasound as a cancer diagnostic tool by selectively binding to EGFR-overexpressing cancer cells for sensitive tumor detection.

## 2. Materials and Methods

### 2.1. Materials

The materials used in the experiments were N-(Methylpolyoxyethyleneoxycarbonyl)- 1,2-distearoyl-sn-glycero-3-phosphoethanolamine (DSPE-PEG-2K, MW 2 kDa, NOF America, White Plains, NY, USA), 1,2-Dipalmitoyl-sn-glycero-3-phosphocholine (DPPC, MW 734 Da NOF America, White Plains, NY, USA), DSPE-PEG-Hydrazide (DSPE-PEG-Hz, MW 3.4 kDa, Nanocs Inc., Boston, MA, USA), chloroform (Oakwood Chemical, Estill, South Carolina, USA), perfluorohexane (Fluoromed, Round Rock, TX, USA), DiD lipophillic fluorescent dye (Biotium, San Francisco, USA), sodium periodate (NaIO_4_, Sigma Aldrich, St. Louis, MO, USA), sodium phosphate dibasic (Na_2_HPO_4_, Sigma Aldrich, St. Louis, MO, USA), in vivo SIM anti human EGFR (Cetuximab Biosimilar) antibodies (Bio X Cell, Lebanon, NH, USA), Alexa Fluor 555 antibody labeling kit (ThermoFisher Scientific, Waltham, MA, USA), Purpald (Alfa Aesar, Tewksbury, MA, USA), BupH coupling buffer (0.1 M sodium phosphate, 0.15 M NaCl, pH 7.2; Thermo Fisher, Waltham, MA, USA), FaDu squamous cell carcinoma cells (ATCC, Manassas, VA, USA), Dulbecco’s Modified Eagle’s Medium (DMEM; Corning, Tewksbury, MA, USA), 10% fetal bovine serum (Gibco-Thermo Fisher, Waltham, MA, USA), 1% penicillin (Corning, Tewksbury, MA, USA), Acrylamide/Bis-acrylamide, 30% solution (Sigma Aldrich, St. Louis, MO, USA), ammonium persulfate (Sigma Aldrich, St. Louis, MO, USA) and Tetramethylethylenediamine (TEMED; Alfa Aesar, Haverhill, MA, USA).

### 2.2. Perfluorohexane Nanodroplet Synthesis

The perfluorocarbon nanodroplets were synthesized using a modified sonication-based method described in Hannah, et al. [27]. A volume of 2.2 µmol of lipids consisting of a 1:0.2:0.03 molar ratio of DSPE-PEG-2k, DPPC and DSPE-PEG-Hz, respectively were added to 1 mL of chloroform in a 50-mL round-bottom flask. The solution was evaporated to form a lipid cake using a rotary evaporator (Heidolph, Schwabach, Germany) at 38.5 °C, 250 mbar and at a speed of 50 rpm. For the nanodroplets used in the fluorescence experiments, 20 µL of 1 mg/mL DiD fluorescent dye was also added to the mixture prior to evaporation.

Upon complete evaporation of the chloroform, the lipid cake was resuspended in 1 mL of deionized water (DI water). The solution was then vortexed (VWR, Radnor, PA, USA) for 30 s and sonicated using a 35-kHz ultrasound water bath sonicator (VWR Symphony, Radnor, PA, USA) at room temperature for 1 min. The resulting mixed lipid solution was transferred to a 20-mL centrifuge tube in an ice bath and 50 µL of perfluorohexane was added to the solution. The mixture was then sonicated using a microtip probe sonicator (QSonica, Newtown, CT, USA) at two different intensities, one of low intensity (1% power, 1 s on, 5 s off, 20 total pulses) followed by a high intensity (50% power, 1 s on, 10 s off, 5 total pulses). Following sonication, the nanodroplets were washed three times by centrifugation (MiniSpin, Eppendorf, Hamburg, Germany) at room temperature first at 43 rcf for 60 s keeping the supernatant, followed by 113 rcf for 60 s, once again keeping the supernatant and finally centrifuged again at 822 rcf for 60 s, this time discarding the supernatant and resuspending the pellet in 1 mL of DI Water for the final PFHnDs. The different centrifugation steps were used to remove larger particles and retain the smaller nanodroplets.

Upon centrifuging, the size and zeta potential of the nanodroplets were determined using dynamic light scattering (DLS) (Zetasizer Nano, Malvern, UK). The stock concentration was diluted 100× for all DLS measurements. The nanodroplet concentration was determined using the DLS size data using the built in concentration calculator. DLS and zeta potential measurements were taken before and after antibody conjugation to the PFHnDs.

### 2.3. Antibody Conjugation to the PFHnDs

EGFR antibodies were conjugated to the surface of the PFHnDs as outlined in Figure 1. First, the EGFR antibodies were filtered using a 30-kDa MWCO centrifuge filter (Sigma Aldrich, St. Louis, MO, USA) for 15 min at 1610 rcf (Sorvall ST16, Thermo Fisher) resulting in a final concentration of 1 mg/mL. The antibodies were then fluorescently stained using the Alexa Fluor 555 (AF 555) antibody labeling kit following the included protocol. The labeled antibodies were stored as a stock solution at 4 °C until needed.

The procedure for oxidizing the Fc region to produce aldehyde groups for binding was adapted from previous work on noble metal nanoparticles [28] and barium titanate nanoparticles [29]. To briefly summarize, 10 µL of the AF 555 labeled antibodies were added to 130 µL of 100 mM Na_2_HPO_4_. 50 µL of this solution was added to 5 µL of 100 mM NaIO_4_. It is important to note that the NaIO_4_ should be made immediately before use for optimal results. After 30 minutes of incubation with NaIO_4_, 150 µL of PBS was added to stop the reaction. Aldehydes were confirmed using the Purpald (Alfa Aesar, Tewksbury, MA, USA) test: 20 µL of the antibody solution was added to 60 µL of a 10-mg/mL solution of Purpald mixed with 1-M NaOH. The purpald solution turns violet in the presence of aldehyde groups.

To attach the antibodies to the surface of the PFCnDs, a directional conjugation strategy was employed using the aldehydes on the Fc region of the antibody to bind to the free hydrazides on the surface of the PFHnDs, creating a stable hydrazone bond. 200 µL of the aldehyde activated antibodies were then added to a solution with 200 µL PFHnDs ($2.2\times 10^{8}$ particles/µL) and 100 µL of BupH coupling buffer (0.1 M sodium phosphate, 0.15 M NaCl, pH 7.2). The mixture was incubated for 4 h at room temperature in the dark on a shaker (10 rcf; Belly Dancer, IBI Scientific, IA, USA). Post incubation, the solution was washed three times at 5433 rcf for 2 min (MiniSpin Centrifuge, Eppendorf, Hamburg, Germany) to remove the unbound antibodies by removing the supernatant each time and resuspending the pellet in 400 µL of PBS. The resulting antibody-nanodroplet product (PFHnD-Ab) was resuspended in 200 µL of PBS after the third wash. Additionally, the amount of antibody conjugated to the nanodroplet surface was quantified using a spectrofluorometer (FluoroMax, Horiba, Kyoto, Japan).

### 2.4. Molecular Targeting of EGFR-targeted PFHnDs to FaDu Cells

FaDu cells (ATCC) were used as the in-vitro model of head and neck squamous cell carcinoma due to their overexpression of EGFR on the surface, which enables molecular targeting via the anti-EGFR antibody. The cells were cultured in T-75 flasks by using DMEM supplemented with 10% fetal bovine serum and 1% penicillin. FaDu cells were suspended in 2 mL of DMEM for a final concentration of $4.4\times 10^{5}$ cells/mL for use in the targeting experiments. Then, 200 µL of the cell solution was treated with 100 µL of $4\times 10^{7}$ nanodroplets/µL solution of fluorescent PFHnD-Abs and incubated for 30 min at 36 °C.

Two control experiments were conducted. First, cells were incubated with PFHnDs without conjugated antibodies to determine the level of non-specific binding. Second, cells were first subjected to unlabeled free floating anti-EGFR IgG (71 µg/mL) to block the specific receptors, inhibiting any EGFR-mediated PFHnD-Ab binding, followed by the addition of the same concentration and volume of PFHnD-Abs as the positive group. To remove unbound nanodroplets, the cells were washed five times for 30 s at 43 rcf (Minispin, Eppendorf), each time removing the supernatant and resuspending in 400 µL of PBS. After the final wash, the cells were then re-suspended in 200 µL of PBS and imaged under a microscope (DMi8, Leica, Germany). Dark-field was used to image the cells, the red fluorescence channel was used for the PFHnDs (DiD), and the yellow fluorescence channel was used for the antibodies (AF 555). To quantify the PFHnD fluorescence, the dark-field images were segmented in MATLAB using the Image Processing Toolbox (MathWorks, Natick, MA, USA). The coordinates of all of the cells in the dark field image were determined and then used to quantify the fluorescence from the corresponding co-registered fluorescence images. A Lilliefors test determined that the data was not normal, requiring the use of a non-parametric statistical test. Therefore, the Mann-Whitney U-test was used to determine the statistical significance of the results.

### 2.5. Effects of Activation on PFHnD-Ab Targeting Efficiency

To determine the effects of activation on PFHnD-Ab targeting efficiency, two experiments were carried out. First, the PFHnD-Abs were exposed to either 0, 10, 100, or 1000 vaporization-recondensation cycles, followed by incubating the nanodroplets with cells. After incubation, the cells were washed using the same procedure and fluorescently imaged to determine the mean nanodroplet fluorescence per cell. Second, in a separate experiment, the cells were incubated with the PFHnD-Abs first, followed by either 0, 10, 100, or 1000 vaporization-recondensation cycles to determine the binding strength of the PFHnD-Abs to cells. Fluorescent images were taken before and after the HIFU activation and the mean nanodroplet fluorescence signal in the cells was compared. A Lilliefors test determined that the data was not normal, requiring the use of a non-parametric statistical test. Therefore, the Mann-Whitney U-test was used to determine the statistical significance of the results.

### 2.6. Ultrasound Imaging of FaDu Cells

To validate the ability of ultrasound to image the PFHnD-Abs targeted to the cells, tissue-mimicking polyacrylamide phantoms were fabricated. Polyacrylamide phantoms have tissue-like acoustic properties such as density, sound speed, acoustic impedance, and attenuation [30,31,32]. A volume of 10 mL of 30% polyacrylamide (PA) solution was added to 20 mL of DI water and 30 µL of 10% *w*/*v* ammonium persulfate to generate a 30-mL phantom. 27 mL of the solution was set into the mold first followed by adding 33.75 µL of Tetramethylethylenediamine (TEMED) crosslinker to create the phantom base layer. The phantom was allowed to set for 5 minutes until hardened. The remaining 3 mL of PA solution was mixed with cells and 3.75 µL of TEMED crosslinker. The solution was added to the top of the base layer and allowed to set.

The experimental setup, ultrasound imaging acquisition sequence, and image processing was adapted from previous work [16]. To briefly summarize, the custom ultrasound imaging setup (Figure 2) used a 15-MHz, 256-element linear array ultrasound transducer (L22-8v, Verasonics, Kirkland, WA, USA) powered by a Verasonics Vantage 256 (Verasonics, Kirkland, WA, USA) ultrasound imaging system and single element high intensity focused ultrasound (HIFU) transducer (H-151, Sonic Concepts, Bothell, WA, USA). A polyacrylamide coupling cone was used to focus the HIFU waveform to the activation spot. The ultrasound image acquisition sequence was synchronized with the HIFU through a digital trigger output from the Verasonics imaging system. The HIFU waveform was generated using an arbitrary waveform generator (Tektronix, Beaverton, OR, USA), which consisted of a 10-cycle, 1.1-MHz sinusoid burst. The waveform was then amplified by a 200-W radio frequency power amplifier (1020 L, E, and I) and sent through an impedance-matching circuit before powering the transducer.

The image beamforming and processing methods were adapted from previous work [16]. The image reconstruction and data processing were completed using the Verasonics reconstruction algorithms. Briefly, each B-mode image was acquired using 5 plane-wave transmissions at angles of −18°, −9°, 0°, 9°, and 18°. Six B-mode ultrasound images were acquired before the first HIFU pulse to be able to reliably determine the background signal. Then, two frames were acquired 500 µs and 400 ms after each HIFU pulse. A delay of 500 µs after the second frame was applied before transmitting the next HIFU pulse. A total of 5 HIFU pulses were applied in each imaging sequence. A differential of frames between pre-HIFU and the first frame after HIFU exposure were used to isolate the signal from the gaseous PFHnDs. The differential ultrasound signal was determined within the 22.5 mm^2^ focal spot region.

### 2.7. Determining the PFHnD-Ab Detection Limits

Cells with attached PFHnD-Ab were diluted 10×, 50×, 100×, and 500× and embedded in a thin layer of polyacrylamide phantoms matching the elevational thickness of the imaging transducer as described in Section 2.6. The initial concentration of cells was determined by using an automated cell counter (Luna II, Logos Biosystems, South Korea). The expected cells per focal area was determined by dividing the cells in the phantom by the area of the HIFU focal zone, 22.5 mm^2^. The phantoms were imaged using the same imaging setup as described in Section 2.6. Each phantom was imaged in five different focal spots (*n* = 5).

### 2.8. Statistical Analysis to Determine PFHnD Ultrasound Contrast

For each acquisition sequence, the first 6 ultrasound frames were averaged to remove noise from the images. The PFHnD signal was isolated from the ultrasound image by taking the difference between the first ultrasound frame after the first HIFU pulse and the averaged background ultrasound frames. The focal spot (6 × 3.75 mm rectangle) was selected as the region of interest (ROI) and all calculations were performed on the ROI. The mean of the ROI was computed for each focal spot (*n* = 5 spots per phantom) and the standard deviation of the ROI was calculated for the group of means. A Lilliefors test determined the data followed a normal distribution. A student’s *t*-test was used to determine statistical significance between the groups. Standard deviation was used to compute the error bars.

## 3. Results

### 3.1. Nanodroplet Characterization and Conjugation

The PFHnD and PFHnD-Abs were characterized for size and zeta potential using DLS, as shown in Figure 3. We obtained a peak size of 534.2 ± 37.5 nm (*n* = 3) and a zeta potential of −12.7 ± 2.01 mV for the non-targeted PFHnDs. After conjugation to the antibodies, the size distribution of the targeted nanodroplets (PFHnD-Ab) increased to a peak of 641 ± 48.5 nm (*n* = 3) and the zeta potential increased to −8.88 ± 3.54 mV (*n* = 3). The as-synthesized concentration was $2.2\times 10^{8}$ nanodroplets/µL as measured by the DLS. The average antibody concentration in the sample was measured to be 1.57 µg/mL. This resulted in an average of $3\times 10^{4}$ antibodies per PFHnD.

### 3.2. Molecular Targeting of PFHnD-Ab to FaDu Cells

The PFHnD-Abs exhibited high levels of binding to the FaDu cells compared to the other groups (Figure 4A) which was confirmed using fluorescence microscopy. The DiD fluorescence signal of the PFHnD-Ab was 5.63× and 6× greater than the signal in the case of non-targeted PFHnDs (without the antibody) and the EGFR blocking control, respectively. The AF 555 fluorescence signal of the PFHnD-Abs was 2.2× and 2.0× greater than the PFHnDs and EGFR blocking control, respectively. The mean fluorescence per cell of the PFHnD-Ab group (*n* = 567 cells) was 42.5 ± 14.5 in the DiD channel and 67.7 ± 19.4 in the AF 555 channel. The control samples averaged 7.6 ± 4.7 in the DiD channel and 30.5 ± 4.1 in the AF 555 channel for the non-targeted group (*n* = 410 cells). The EGFR-blocking group (free floating IgG (*n* = 558 cells) exhibited a mean cellular fluorescence of 7.1 ± 4.0 in the DiD channel and 33.6 ± 9.6 in the AF 555 channel. The cell auto-fluorescence without the presence of any nanodroplets or antibodies was measured to be 5.8 ± 1.0 in the DiD channel and 34.6 ± 4.3 in the AF 555 channel. The PFHnD-Ab sample DiD fluorescence was statistically significant compared to all conditions, as shown in Figure 4B (*p* < 0.001). Some nonspecific binding of non-targeted PFHnDs to the cells was observed; however, there was not any significant statistical difference in the fluorescence with relation to the cell auto-fluorescence.

### 3.3. Effects of Repeated Vaporization of PFHnD-Abs and Cellular Binding

One of the advantages of using perfluorohexane in the core of the nanodroplets is the ability to repeatedly activate the particles from a liquid to gas state. We found that activating the nanodroplets before allowing them to incubate with cells did not impact nanodroplet binding to the cells, confirmed by fluorescent microscopy (Figure 5A) in both the DiD and AF 555 channels. In some isolated cases, there was a significant difference in the mean fluorescence per cell; however, due to a lack of consistent trend, we attribute this to the moderate number of cells imaged (Figure 5B,C). Cells activated with HIFU with PFHnD-Abs already bound to the surface, showed no difference in the mean fluorescence per cell confirmed by fluorescent images before and after HIFU activation (Figure 5D). The mean fluorescence per cells before and after HIFU treatment for the DiD and AF 555 channels are shown in Figure 5E,F respectively.

### 3.4. Ultrasound Imaging of PFHnD-Ab Targeted Cells in Phantoms

We found that the phantom containing cells with the PFHnD-Ab resulted in the highest ultrasound signal compared to the control groups, with a *p*-value < 0.001 as shown in Figure 6. We observed some ultrasound contrast in the other two control groups, PFHnDs and IgG blocking (free antibody + PFHnD-Abs), due to limited non-specific binding and incomplete washing. The ultrasound differential amplitude of the targeted group was statistically greater than the other groups, as shown in Figure 6B. Repeated vaporization of the PFHnD-Abs was confirmed from the captured ultrasound images and quantification of the differential ultrasound amplitude for each HIFU pulses, as shown in Figure 6C. Cells on their own without any nanodroplets did not exhibit any ultrasound contrast, indicating that the HIFU energy levels used in this study did not exceed the cavitation threshold.

### 3.5. Determining the PFHnD-Ab Detection Limits

The initial cell concentration was measured to be $2.12\times 10^{5}$ cells/mL. The average number of cells in the HIFU focal zone was determined by dividing the cells in the phantom by the area of the focal zone, 22.5 mm^2^. This resulted in 38.16, 7.63, 3.81, and 0.76 cells per focal area for the 10×, 50×, 100×, and 500× dilutions, respectively. Interestingly, the more diluted samples showed individual points of signal, indicating single-cell detection (Figure 7A). Overall, the differential ultrasound signal was linearly proportional to the cell concentration, with an *r*^2^ value of 0.998 (Figure 7B).

## 4. Discussion

The PFHnD-Abs developed show promise to augment the use of ultrasound molecular imaging in cancer diagnostics. Fluorescent images of cells showed high specific binding of the PFHnD-Abs to the surface of the cells. The ultrasound images also showed high ultrasound contrast for cells with bound PFHnD-Abs. Single-cell sensitivity was showcased in the ultrasound images, implying that not many nanodroplets are needed to produce imaging contrast.

The conjugation strategy employed in attaching the antibodies to the PFHnDs has multiple benefits. First, the aldehyde-hydrazide chemistry is a one-step procedure that does not require specialized linker molecules and can be carried out in biologically appropriate conditions. Second, the method could be adapted to any IgG antibody with a glycosylated Fc region, allowing for conjugation of antibodies on the surface for different molecular targets. Third, the PFHnDs are bound to the Fc region of the antibody, outwardly exposing the Fab binding regions of the antibody, which increases the binding efficiency and reduces the number of antibodies required to achieve effective targeting.

While the size of the PFHnDs developed in this study are sub-micron, further size optimization can be performed by modulating the lipid shell composition, choice of perfluorocarbon core and sonication/washing steps [33,34]. The perfluorohexane core used in this study allows for repeated vaporization and recondensation after undergoing acoustic droplet vaporization, enabling imaging of the same nanodroplets for improved contrast. The higher boiling point perfluorohexane core also allows for better stability and a lower likelihood of spontaneous vaporization [15,16,27]. Importantly, the results in this paper show that the PFHnD-Ab retain their molecular targeting capabilities even after exposure to 1000 HIFU pulses. This suggests that the PFHnD-Ab have the promise to be robust in-vivo contrast agents for molecular ultrasound imaging.

Although the perfluorohexane core results in less volatile nanodroplets compared to other perfluorocarbons, it also requires a relatively large amount of energy to initiate the vaporization. Previous studies have relied on the use of a pulsed laser for activation to realize the necessary energy threshold. Optical penetration limits the imaging depth in tissue [15,27,35]. The use of HIFU to activate the nanodroplets allows for deep-tissue activation [16]. The HIFU pressure levels used in this study were below the cavitation threshold, and no signs of cavitation were observed; however, the biological effects of HIFU in combination with PFHnDs has not yet been determined in vivo.

Importantly, the conjugation strategy does not rely on the use of perfluorohexane. It could be applied to the more volatile perfluoropentane (b.p. 28 °C) or perfluorobutane (b.p. 4 °C).These nanodroplets would require less energy for activation (i.e., a conventional imaging transducer could be used), but would only offer a single-time vaporization. Thus, more care would need to be taken to ensure binding in a region prior to imaging. An additional benefit of using a lower-boiling-point core is the ability to release cargo on demand. Therefore, the nanodroplets could be used for molecularly targeted drug delivery with ultrasound image guidance [18,36,37].

Overall, the work presented here can be expanded to a platform of different ultrasound contrast nanodroplets through the use of other antibodies and different PFC cores. The activation and imaging system previously developed [16] works well in conjunction with the developed nanodroplets to expand the capabilities of molecular ultrasound imaging. Potential benefits to current diagnostics include faster imaging, earlier detection, lower costs, less invasiveness, and zero exposure to radiation compared to other imaging modalities such as MRI, PET, and CT, as well as biopsies.

## 5. Conclusions

In this study, we have presented a novel EGFR-targeted ultrasound imaging contrast agent for improved molecular imaging. The conjugation of EGFR antibodies onto the surface of the PFHnDs showed strong association with EGFR-overexpressing FaDu cells with minimal nonspecific binding. We also demonstrated that the particles bound to cells could be imaged with a linear array ultrasound transducer after using HIFU to activate the PFHnDs. These findings show that the PFHnD-Abs are a robust contrast agent that could be applied to detect EGFR-overexpressing cells.

## Figures and Tables

**Figure 1 nanomaterials-12-02251-f001:**
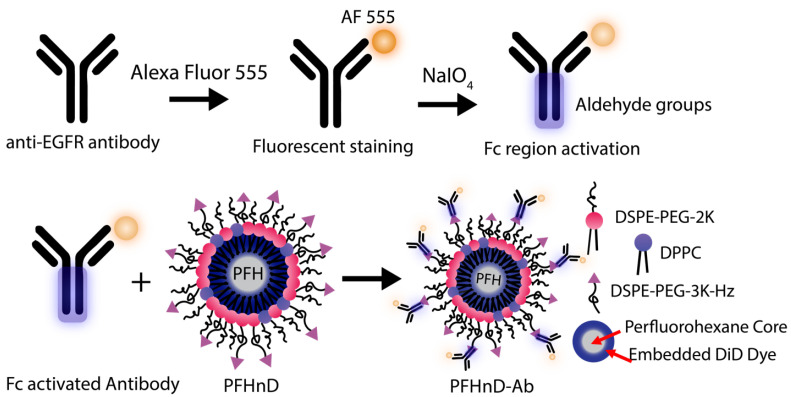
The antibodies were first labeled with AF 555 antibody labeling kit. Second, the Fc region of the antibody was oxidized by reacting with 100-mM NaIO_4_ leading to aldehydes on the Fc region. The aldehydes on the activated Fc region then bind to the hydrazides available on the surface of the PFHnDs, forming a stable hydrazone bond and leading to the final antibody conjugated nanodroplet (PFHnD-Ab). The core of the PFHnD contains perfluorohexane (b.p. 56 °C) and DiD fluorescent dye.

**Figure 2 nanomaterials-12-02251-f002:**
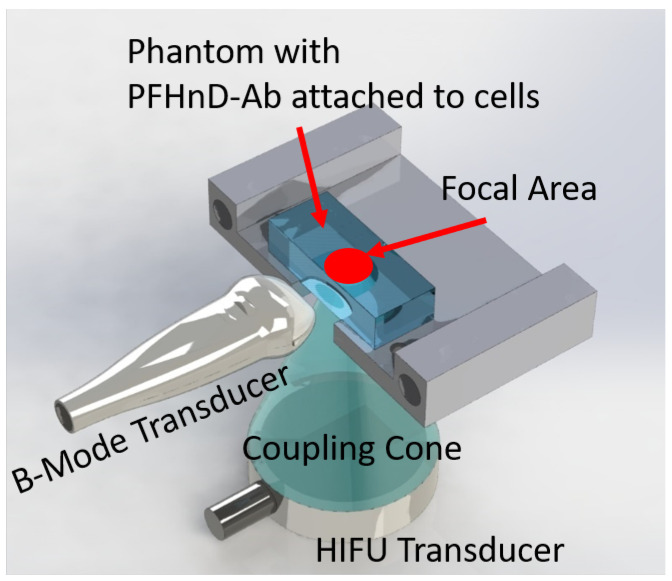
Schematic of the custom ultrasound setup. The single element HIFU transducer uses a polyacrylamide coupling cone to deliver the activation energy needed to induce the PFHnD-Ab phase change. Polyacrylamide phantoms containing PFHnD-Abs conjugated to cells were placed on the 3D printed stage. A linear array B-mode transducer was placed perpendicular to the activation plane to capture the vaporization and recondensation processes.

**Figure 3 nanomaterials-12-02251-f003:**
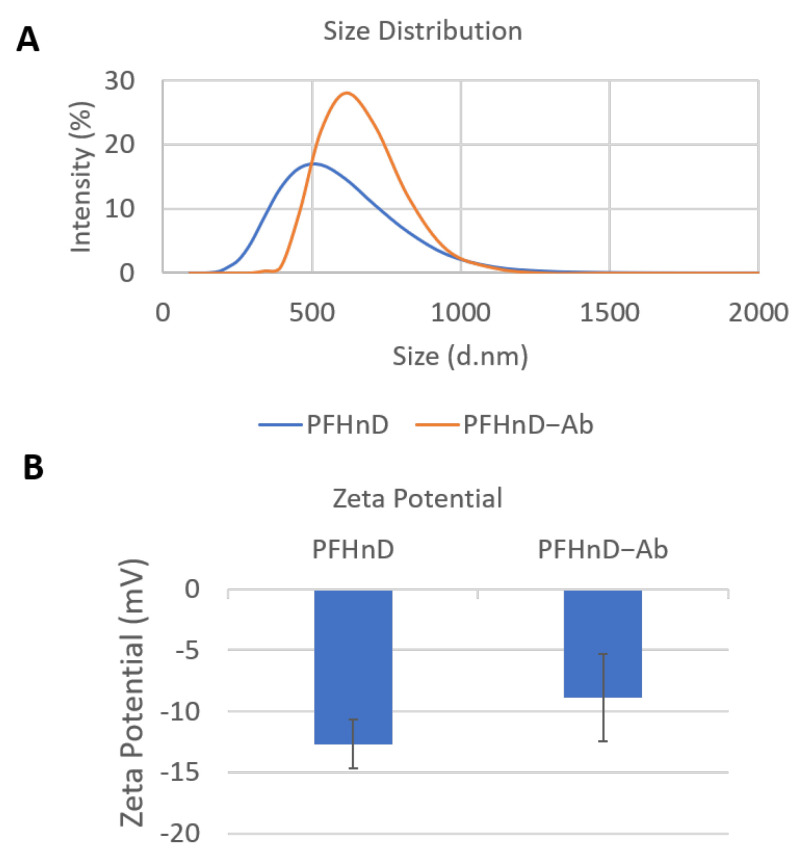
(**A**) The size distribution of both the non-targeted PFHnDs (534.2 ± 37.5 nm; *n* = 3) and antibody conjugated PFHnD-Ab (641 ± 48.5 nm; *n* = 3). (**B**) The zeta potential of both targeted (PFHnD-Ab; −8.88 ± 3.54 mV; *n* = 3) and non-targeted (PFHnD; −12.7 ± 2.01 mV; *n* = 3) nanodroplets.

**Figure 4 nanomaterials-12-02251-f004:**
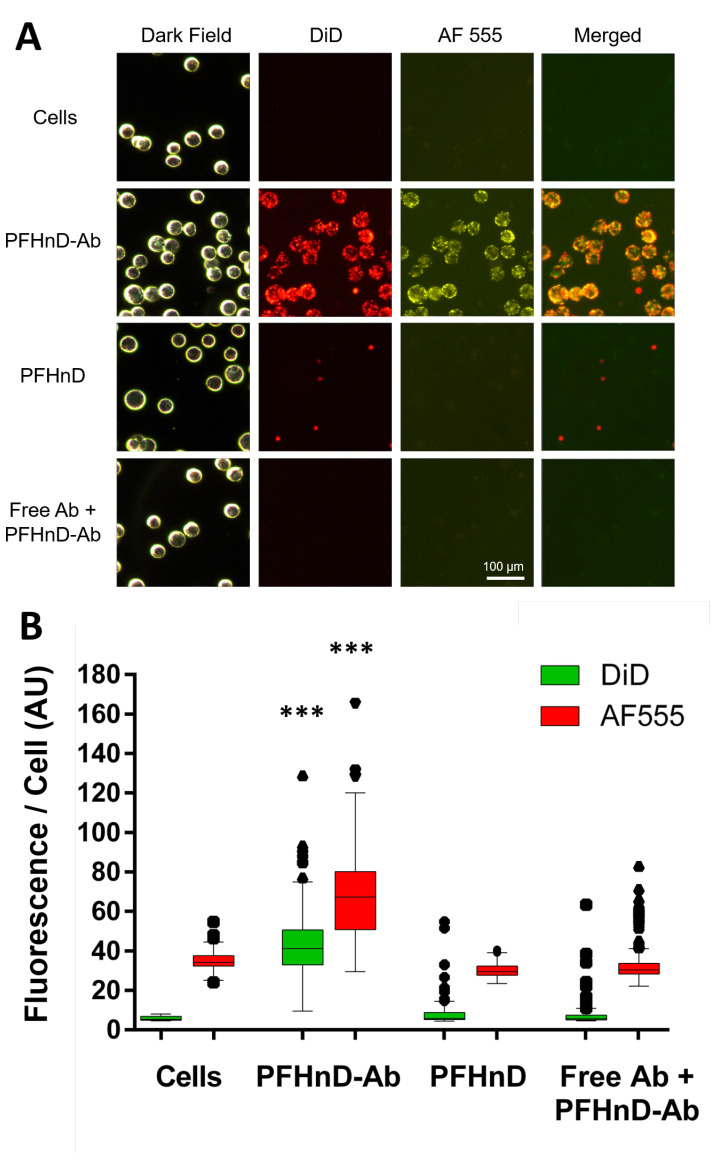
(**A**) FaDu cells were imaged using a fluorescent microscope for the different groups; cells, cells with targeted PFHnD-Abs, cells with non-targeted PFHnDs, and cells that were exposed to unlabeled free antibody prior to incubating with targeted PFHnD-Abs (blocking study). DiD lipophilic dye was embedded into the PFHnD-Ab and PFHnDs, while the antibodies were fluorescently labeled with AF 555. Dark-field cell images were used to identify the location of each cell in the image. The locations were then used to compute the fluorescent signal within the cell. Merged images show the two fluorescent channels (DiD and AF 555) overlaid. (**B**) The mean DiD and AF 555 fluorescence per cell for each group was computed from the fluorescent images, resulting in the PFHnD-Ab group having a statistically significant (*** = *p* < 0.001) higher mean DiD and AF 555 fluorescence/cell compared to the all other groups. There was not any statistical significance between the control groups.

**Figure 5 nanomaterials-12-02251-f005:**
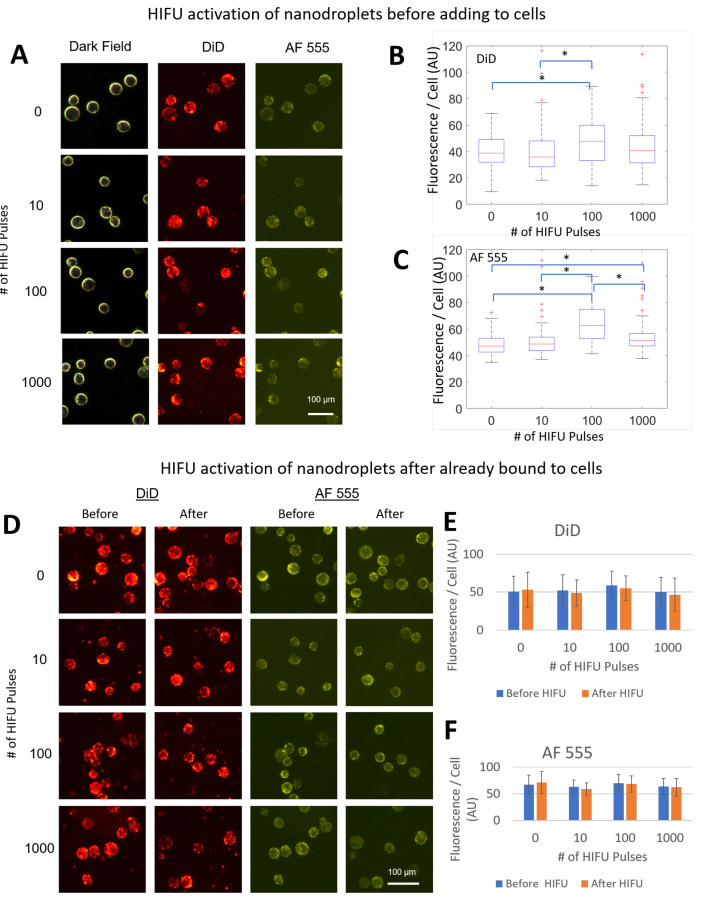
The PFHnD-Abs were subjected to either 0, 10, 100, or 1000 vaporization-recondensation cycles using HIFU before being incubated with cells to test the effects of activation on cellular binding. (**A**) Fluorescent images of cells after incubation with the pre-activated PFHnD-Abs. (**B**) The mean fluorescence per cell in the DiD channel showed no statistically significant difference except for two isolated cases (* *p* < 0.01) using the Mann-Whitney Test. (**C**) The mean fluorescence per cell in the AF 555 channel resulted in isolated cases of statistically significant differences using the Mann-Whitney test (* *p* < 0.01). (**D**) Before and after images of cells with PFHnD-Abs bound to the surface before HIFU activation in the DiD and AF 555 channels. (**E**,**F**) The mean fluorescence per cell before and after HIFU activation did not show any statistical difference in the DiD and AF 555 channel respectively.

**Figure 6 nanomaterials-12-02251-f006:**
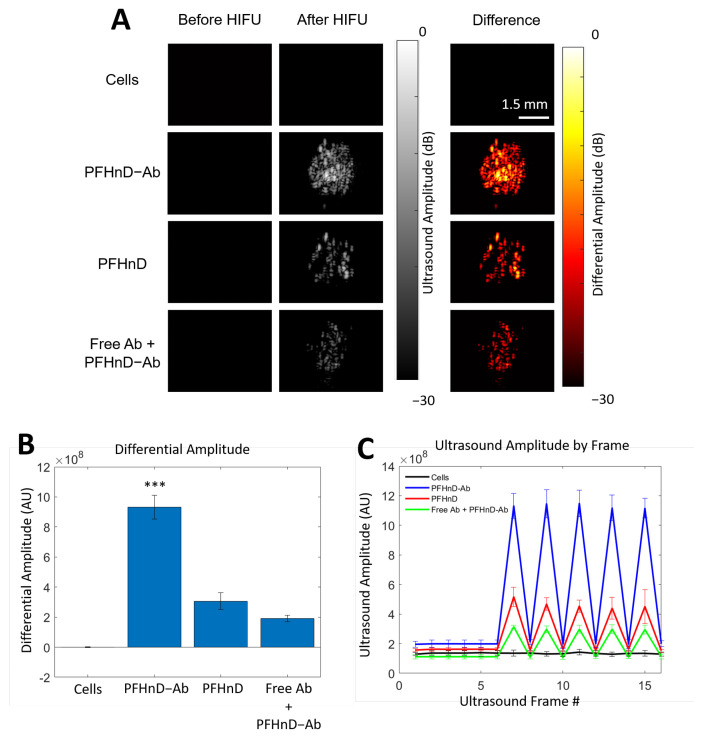
Ultrasound images of cells embedded in polyacrylamide tissue-mimicking phantoms before and after the HIFU activation of the PFHnDs. (**A**) The difference between the two images is displayed in the right panel, indicating PFHnD activation. Cells conjugated with the targeted droplets (PFHnD-Ab) were imaged along with the control groups; non-targeted droplets (PFHnD), blocking group (Free antibody and PFHnD-Ab), and FaDu cells without PFHnDs. (**B**) The differential amplitude was calculated for each of the groups and resulted in the PFHnD-Ab group having a significantly higher differential amplitude (*** *p* < 0.001) than the other groups. (**C**) The average ultrasound amplitude within the focal spot for each ultrasound frame captured for each of the groups. The HIFU was pulsed prior to frames 7, 9, 11, 13, and 15, resulting in the ultrasound amplitude spike within those frames.

**Figure 7 nanomaterials-12-02251-f007:**
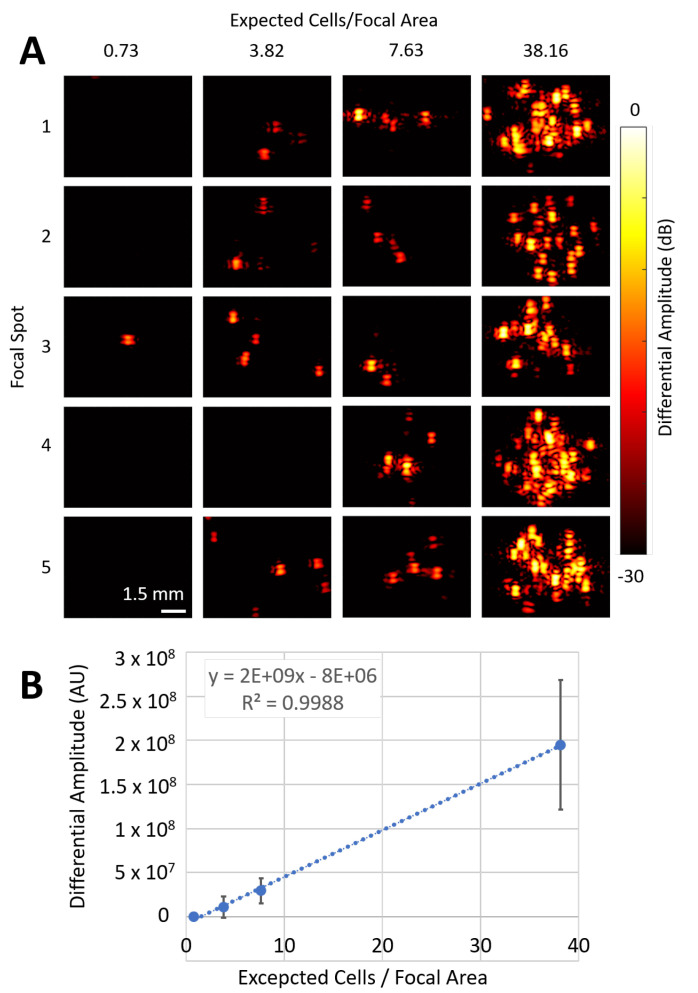
Cells with PFHnD-Abs attached to them were diluted to different concentrations and embedded into polyacrylamide phantoms. (**A**) Images of the differential ultrasound amplitude for each of the different concentrations. The expected cells/focal area was calculated based on the initial concentration of cells. (**B**) The differential amplitude plotted as a function of the expected cells/focal area yielded an *r*^2^ of 0.998.

## Data Availability

The data presented in this study are available on request from the corresponding author.

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
