# Peer review of "EGFR-Targeted Perfluorohexane Nanodroplets for Molecular Ultrasound Imaging"

_nanomaterials, 2022, doi:10.3390/nano12132251_

Round 1
Reviewer 1 Report
Please double check your article and make further revisions.

Reviewer 2 Report
The ms presented by Jandhyala et al focuses on ultrasound imaging imaging for cancer diagnostics. They use nano sized perfluororcarbon (PFCnDs) entrapped in lipids vesicles as ultrasound contrast agents. Perfluorocarbons provide an imaging contrast due to the large acoustic impedance mismatch between their gaseous core and the surrounding tissue and the ultrasound contrast is produced through multiple vaporization/recondensation cycles PFCnDs within the lipid vesicle/droplet . Authors describe as a proof-of-principle, a series of in vitro ultrasound imaging experiments of anti-EGFR antibody-functionalized PFCnDs, to then apply this at a later stage, for cancer diagnostics.of HNSCC (head and neck squamous cell carcinomas). They tested the impact of repeated vaporization cycles on molecular binding (EGFR antibody recognition) on FaDu cells (cell model for squamous cell carcinoma derived from a male patient) and then optimized probe concentration within the FaDu cells for optimal imaging, reaching single cell concentrations. Overall, the experimental setup is straight forward and sound and the presented results could be optimized here and there, but have a clear message with respect to the main objective of the study.
It would have been nice to see different cell models (cancer and non cancer cells),that could have given some data on the general imaging noise to validate general aspect of this approach. It also would have been interesting to see or discuss whole tissue (not single unilayer cells) to get an idea of the strength and potential of this imaging approach. Also the use of different antibodies (for different types of cancer) would have given a more general information and supported the use of this techique for cancer diagnostics.
Authors should also discuss their approach to existing diagnostics (costs vs benefits; sensitivity; early, fater and better disgnostics might allow a less invasive cancer treatment).
Though I think its is a very short comunication, which would benefit from more general studies to validate it for cancer diagnostics I have no objection to recommend it for publication.
Minor issues:
Figure 4, Show a zoomed merged image of the droplets. This is not important for diagnostics as it will probably be done with a cell sorter conneted device, but it will improve the Figure
Figure 5 B and C: please set the same Y axis, this will make it easier to compare data set as no numbers are given in the text.
Legend Figure 5: A) Fluorescent images of cells after incubation with the pre-activated PFHnD-Abs are shown in Figure 5A.
"are shown in Figure 5A." is redundant as you are telling us what you show in Figure 5A….
Reviewer 3 Report
This paper adds to the current research of this type of microbubbles which are termed by the authors as nano-droplets" though their sizes are in microns.
It is well written and well referenced.
The authors modified slightly the currently investigated agent :Perfluorocarbone".
The main issue with this research is that fact that the authors are claiming that this can be used in ultrasound imaging of patients. However, they are using different type ultrasound. They need to show that the same effects can be obtained in using typical ultrasound probes used in medicine for imaging.
If they intend to use this technique for cells study, then they have to prove that this technique is superior to the other bio-technological methods used in labs.
Therefore, I suggest that the authors need to explain these issues first before considering the manuscript for publication.
Round 2
Reviewer 3 Report
Please see attachment.

Author Response
Please see the attachement
